# Fecal and skin microbiota of two rescued Mediterranean monk seal pups during rehabilitation

Aggeliki Dosi,[1] Alexandra Meziti,[1] Eleni Tounta,[2] Kimon Koemtzopoulos,[2] Anastasia Komnenou,[3] Panagiotis Dendrinos,[2] Konstantinos Kormas[1]

**ABSTRACT** The role of animal host-associated microbiomes is becoming more apparent and defined for wild animals, especially for the species under conservation strategies. This study investigated the succession of fecal and skin bacterial microbiota of two rescued female Mediterranean monk seal (*Monachus monachus*) pups for most of their rehabilitation period. Bacterial species richness and diversity were assessed by high-throughput sequencing of nine freshly collected fecal samples and four skin swabs per individual. Both the fecal and skin microbiota highly overlapped in their containing operational taxonomic units (OTUs) and abundance patterns. The fecal microbiota was separated in two distinct periods and was dominated by OTUs related to the *Shigella*, *Streptococcus*, *Enterococcus*, *Lactobacillus,* and *Escherichia* genera in the first period, while in the second period, the dominating genera were the *Clostridium, Blautia, Fusobacterium, Edwardsiella,* and *Bacteroides*. The skin microbiota was highly similar between the two individuals in each sampling and was dominated by *Psychrobacter-, Elizabethkingia-,* and *Bergeyella*-related OTUs. The provided antibiotic treatment, along with the provided probiotics and nutritional supplements, resulted in a major turnover of the bacterial microbiota with the potentially detrimental OTUs being eliminated toward the end of the rehabilitation period, prior to the release of the pups in the wild.

**IMPORTANCE** This study showed that during the rehabilitation of two rescued Mediterranean monk seal pups (*Monachus monachus*), the skin and fecal bacterial communities showed similar succession patterns between the two individuals. This finding means that co-housed pups share their microbiomes, and this needs to be considered in cases of infection outbreaks and their treatment. The housing conditions, along with the feeding scheme and care protocols, including the admission of antibiotics as prophylaxis, probiotics, and essential food supplements, resulted in bacterial communities with no apparent pathogenic bacteria. This is the first contribution to the microbiome of the protected seal species of *M. monachus* and contributes to the animal's conservation practices through its microbiome.

**KEYWORDS** conservation, marine mammal, red list, threatened, *Monachus monachus*, host–microbe interaction

The introduction of the holobiont and hologenome concepts (1) intrigued the scientific interest in the investigation of the associations and interactions between wild animals and their microorganisms (2, 3). This research promotes the notion that the animals as hosts have inseparable evolutionary and functional roles with their microbiomes (4). Recently, the concept that our view on animals should shift from the object (animal organism) to the process (animal organisms + its everchanging microbiome) ontology has been proposed (5). Such tight and vital symbiotic relationships between macroorganisms and their microorganisms are pivotal for the host's health,

Address correspondence to Konstantinos Kormas, kkormas@uth.gr.

The authors declare no conflict of interest.

development, and nutrition, and for this, there is no reason not to consider them important, especially during ecological disturbances (6). Host–microbe interactions work, among other functions, as buffers against various kinds of biological or environmental disturbances to maintain the host's homoeostasis and function (7). Indeed, similar rescuing roles have been proposed for environmental microbiomes too (8).

It is recently estimated that 42,100 plant and animal species face extinction, with 27% of them being mammals (9). This sets the need for intensifying our current and effective conservation strategies and also to search for more novel approaches. The development of such strategies requires a more holistic knowledge of the organisms under threat. Soon after the first case studies showing the importance of microbiomes in conservation issues of wild animals [e.g., (10, 11)] the conceptual importance of animal microbiomes was revealed (12, 13). The field has progressed so fast that even well-established human microbiome-based manipulations for therapeutic purposes are now applied for captive animals to improve the animals' health (14). Regarding animal species with both natural (wild) and captive (zoos, laboratory animals, farmed species, etc.) populations, the comparison between the two types of microbiomes is considered as the first important step toward knowing the animals' natural microbiome and/or the impact of biological or environmental disturbances (15–18). More specifically, for endangered animals that are under targeted protection actions, microbiome analysis is often restricted to only captive animals' hospitalization or rehabilitation of individuals from the wild for enhancing the species natural population.

Despite the fact that the total biomass of marine mammals across the world's oceans is only 0.3% of the total marine animal biomass, when arthropods and fish have 38.6% and 27.0% (19), respectively, several of these species are endangered and protected due to anthropogenic activities, like the Mediterranean monk seal, *Monachus monachus* (20). More than half of the estimated total marine mammal biomass (ca. 40 Mt) is attributed to baleen whales, leaving seals with a low biomass contribution (21). However, apart from being the most endangered pinniped species in the world, *M. monachus*, the only species of the *Monachus* genus (22, 23), has several other points that attract scientific interest. Although once abundant throughout the Mediterranean, in the Black Sea, and in the northwest Atlantic coast of Africa, the Canary Islands, the Azores, and the Madeira archipelago, its distribution is now limited to the eastern Mediterranean Sea, an isolated population in the Atlantic coast of Africa (Mauritania), and a small isolated population in Madeira due to human-induced population declines through persecution and habitat destruction (24). Today, the species is designated as "endangered under criteria D" by the International Union for Conservation of Nature List of Threatened Species (25) with signs of improving population through targeted scientific research, monitoring of local seal populations, education, public awareness and citizen science campaigns, and rescue and rehabilitation of wounded, sick, and orphaned seals (24, 26, 27). The latter includes animal care practices on land for both young and adult individuals with specific medical and targeted nutritional care. *M. monachus* has been recently recognized as one of the Mediterranean marine mammals' species and a highly vulnerable and high-climate-risk species (28), and this might impose additional burden on the animal's care efforts in the future.

To date, there is no available scientific literature on any *M. monachus* microbiota or microbiome, although it has been shown that the Phocidae family seems to clearly differentiate in their fecal microbial communities compared to terrestrial carnivores (29). The only available relevant knowledge stems from its closely related species, the Hawaiian monk seal (*Neomonachus schauinslandi*), and refers to the identification of either bacterial antibodies of hauled-out specimens (30), cultured aerobic bacteria of the upper respiratory tract of captive animals (31), and bacterial pathogen prevalence in the blood serum of experimentally resident and translocated seals (32). One reason for such restricted knowledge is related to the multiple challenges raised in microbiome sampling from marine mammals. However, several of these marine mammalian species are hospitalized, rehabilitated, or live in captivity in zoos where their microbiomes,

although far from that of their natural counterparts, are far more reachable. In this paper, we report, for the first time, on the fecal and skin microbiota succession of two rescued female *M. monachus* pups (named as Lena and Nicole) during their 5-month rehabilitation period prior to their release in the wild. We evaluated the impact of the provided medication and feeding scheme on these microbiota profiles by analyzing the changes in the skin and fecal bacterial community composition.

## RESULTS

After applying quality filtering and chimera removal of the 16S rRNA gene V3–V4 region amplicons, a total of 807,391 sequences were retrieved. The number of sequences per sample was rarefied to be equal to the smallest number (29,817) of sequences per sample. These sequences were assigned to 328 unique operational taxonomic units (OTUs) at a similarity cut-off level of 97%. The dominant bacterial phyla in the whole data set were Firmicutes, Gammaproteobacteria, Bacteroidota, Fusobacteria, and Actinobacteria.

On average, Lena had 189 ± 12.0 and 244 ± 11.0 OTUs in her fecal and skin samples, while the respective values for Nicole were 288 ± 12.4 and 234 ± 7.4 (Fig. 1; Tables S1 and S2). The Simpson 1-*D* diversity index showed little variance between the two individuals, as it averaged 0.85 ± 0.068 and 0.87 ± 0.026 in Lena's and Nicole's feces, respectively, while in their skin samples, it averaged 0.21 ± 0.024 and 0.93 ± 0.021, respectively. The two individuals shared 88.8% of their fecal and 89.4% of their skin total OTUs (Fig. S1). Permutational multivariate analysis of variance (PERMANOVA) showed that for each pup, skin and fecal bacterial microbiota were significantly different, but no such differences occurred between the two pups' skin and fecal microbiota (Table 1). However, the fecal bacterial microbiota was significantly different from the skin bacterial microbiota in both individuals (Table 1).

Cluster analysis of the feces bacterial microbiota, based on Bray–Curtis similarity, revealed two major groups corresponding to two time periods, with similarity of ≤40% (Fig. 2). The first group included the first three sampling dates for both pups and was dominated (≥5% relative abundance in the whole period) by OTUs affiliated with the *Shigella* (35.2% relative abundance), *Streptococcus* (17.1%), *Enterococcus* (9.7%), *Lactobacillus* (6.4%), and *Escherichia* (5.5%) genera. The second group included the rest of the samplings. During this period, the most abundant fecal bacteria OTUs were associated with the *Clostridium* (44.5% relative abundance), *Blautia* (15.0%), *Fusobac-*

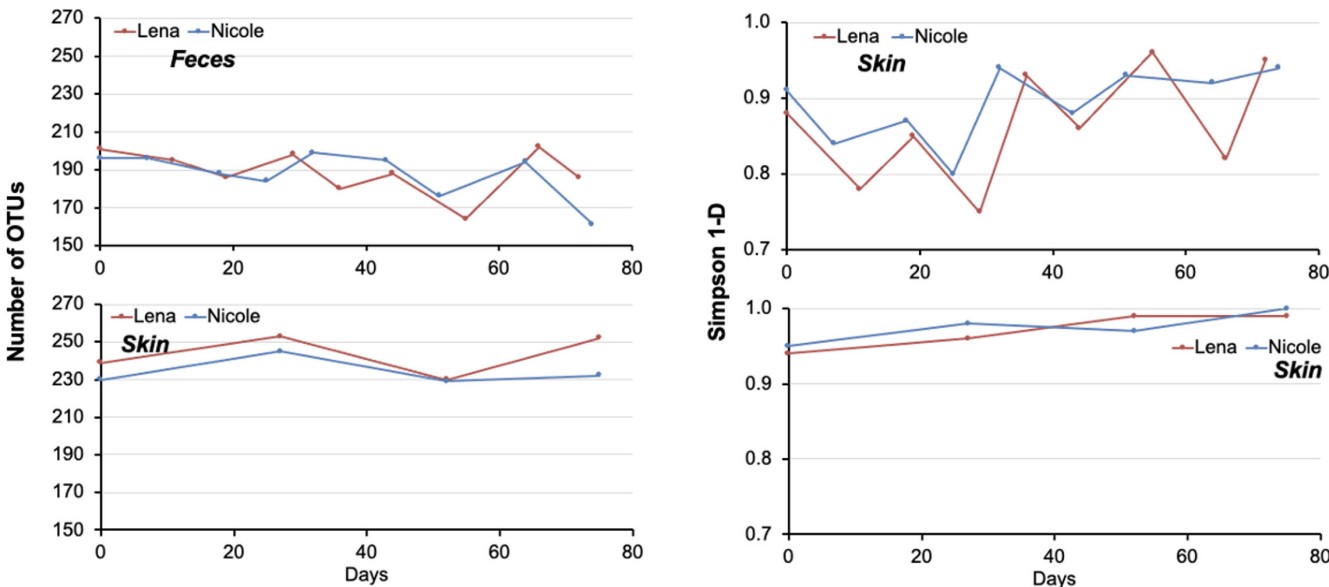

**FIG 1** Bacterial operational taxonomic unit richness and Simpson 1-*D* index in the feces and skin of two hospitalized *Monachus monachus* pups.

**TABLE 1** PERMANOVA results of the fecal and skin bacterial operational taxonomic units in the feces and skin of two hospitalized *Monachus monachus* pups[a]

|  | Lena-feces | Nicole-feces | Lena-skin | Nicole-skin |
|---|---|---|---|---|
| Lena-feces | X | 0.689 | 0.002* | 0.001* |
| Nicole-feces | 0.568 | X | 0.002* | 0.002* |
| Lena-skin | 5.688 | 5.751 | X | 0.797 |
| Nicole-skin | 6.218 | 6.313 | 0.160 | X |

[a]The upper half is the *P*-values, and the lower half of the table is the *F*-values. The star indicates $P < 0.05$.

*terium* (10.3%), *Edwardsiella* (5.5%), and *Bacteroides* (5.3%) genera. For both individuals, statistically significant differences were found in the fecal bacterial communities between consecutive sampling points in most cases (Fig. 2). The dominant fecal OTUs (cumulative relative abundance ≥80%) of both individuals consisted of 46 OTUs (Fig. 3) with 27 of them being shared among the two individuals (Fig. S2).

Regarding the skin bacterial microbiota, four clusters were formed with each one containing the bacterial profiles of both individuals in each sampling point (Fig. 4). In each pair, a *Psychrobacter*-related OTU dominated (44.9%–72.2% relative abundance) while other abundant OTUs were affiliated with the *Bergeyella* (2.2%–20.1%) and *Elizabethkingia* (0.1%– 14.8%). No statistically significant differences were found between any of the individual's pairs, apart from day 27 (Fig. 4). A total of 41 OTUs dominated in both individuals, with 26 of them being shared among the two individuals (Fig. S2). The top dominant OTUs belonged to the *Psychrobacter*, *Bergeyella*, and *Elizabethkingia* (Fig. 5). The *Psychrobacter*-like OTU over-dominated in both individuals in all sampling points, with its higher abundance in the first sampling.

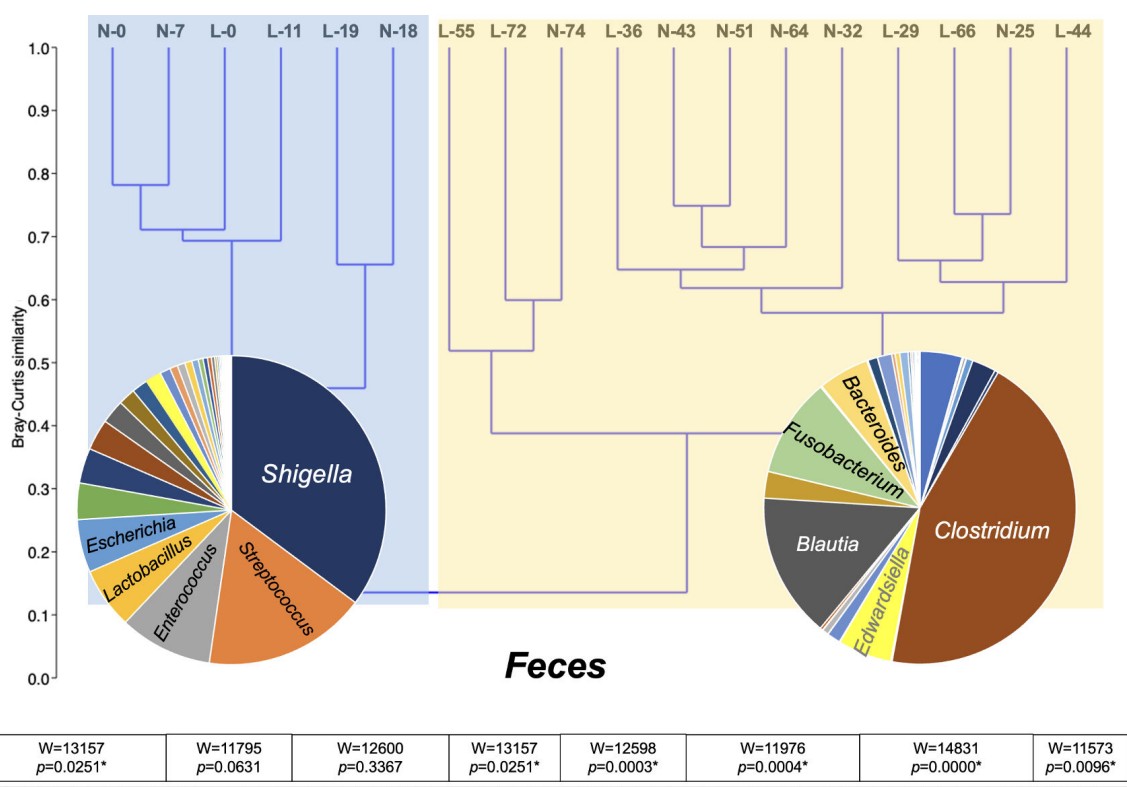

**FIG 2** Cluster analysis of the bacterial operational taxonomic unit abundances in the feces of two hospitalized *Monachus monachus* pups and statistical differences (* indicates $P < 0.05$) between consecutive sampling dates. Shaded areas represent the two major groups (see text). L, Lena; N, Nicole. L/N numbers indicate the day of sampling. Abundances in the pie charts represent the average relative abundance of bacterial genera within each cluster of samples.

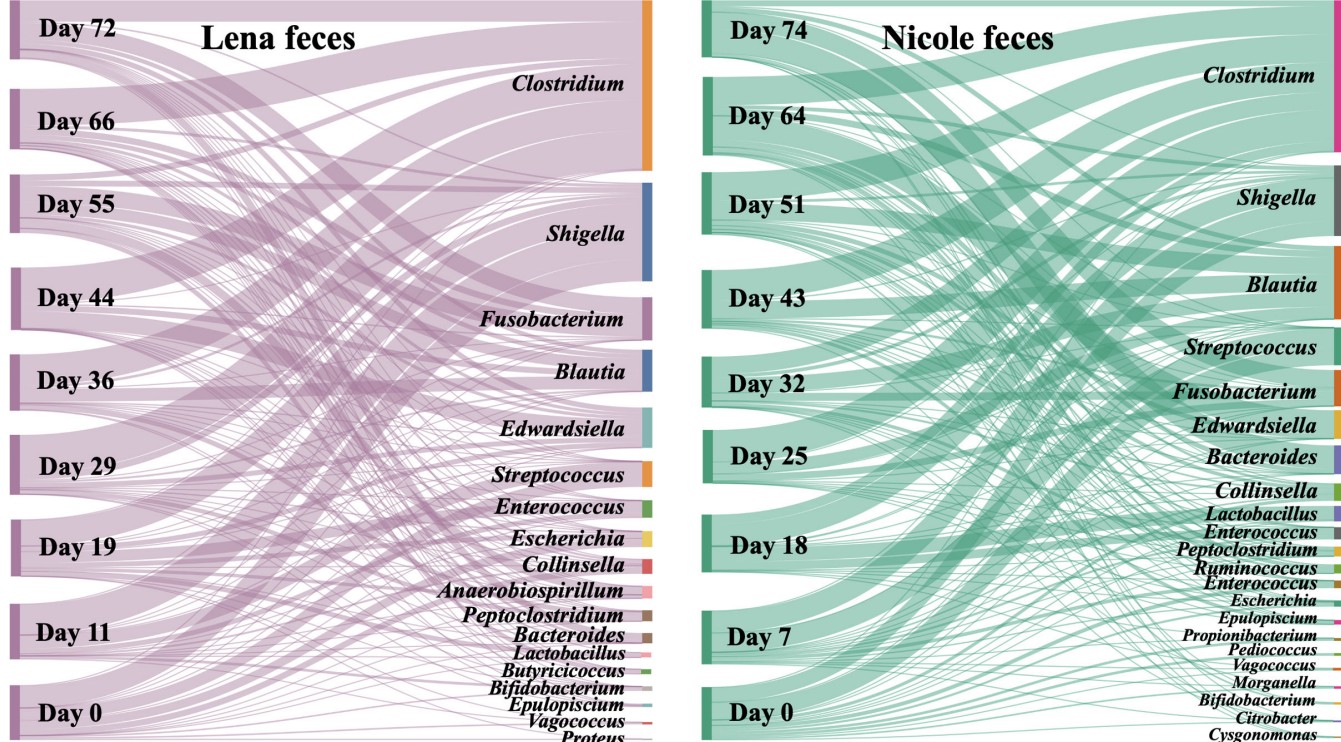

**FIG 3** Dominant (≥80% relative abundance) bacterial genera found in the feces of two hospitalized *Monachus monachus* pups in each sampling point.

The robustness of the Lena fecal bacterial communities increased 1.8 times until day 55 and decreased 1.4 times until the last sampling (Table 2). Nicole's fecal bacterial communities showed a continuous increase in robustness (×3.4) until the last sampling. Regarding the skin bacterial communities, robustness increased 3.0 and 2.5 times until the end of all samplings.

## DISCUSSION

Microorganisms contribute through positive and negative effects to the adaptability and fitness of their animal hosts (13) by taking part in or even regulating processes related to their host's nutrition physiology, reproduction, development, behavior, and susceptibility or resistance to infectious diseases (33). Accumulating scientific literature emphasizes that host-associated microbial consortia occurring from the animals' skin to their gastrointestinal tracts can be informative and supporting or complementary tools to conservation practices (7, 12, 13, 34, 35). Apart from humans, the significance

**TABLE 2** Robustness index ($S_T$) of the fecal and skin bacterial communities found in two hospitalized *Monachus monachus* pups[a]

| Feces | | | | Skin | | |
|---|---|---|---|---|---|---|
| | Lena $S_T$ | | Nicole $S_T$ | | Lena $S_T$ | Nicole $S_T$ |
| 0–11 d | 0.033 | 0–7 d | 0.041 | 0–27 d | 0.092 | 0.109 |
| 11–19 d | 0.032 | 7–18 d | 0.043 | 27–52 d | 0.105 | 0.112 |
| 19–29 d | 0.039 | 18–25 d | 0.044 | 52–75 d | 0.186 | 0.162 |
| 29–36 d | 0.036 | 25–32 d | 0.042 | | | |
| 36–44 d | 0.049 | 32–43 d | 0.056 | | | |
| 44–55 d | 0.067 | 43–51 d | 0.052 | | | |
| 55–66 d | 0.055 | 51–64 d | 0.061 | | | |
| 66–72 d | 0.048 | 64–74 d | 0.141 | | | |

[a]d, day.

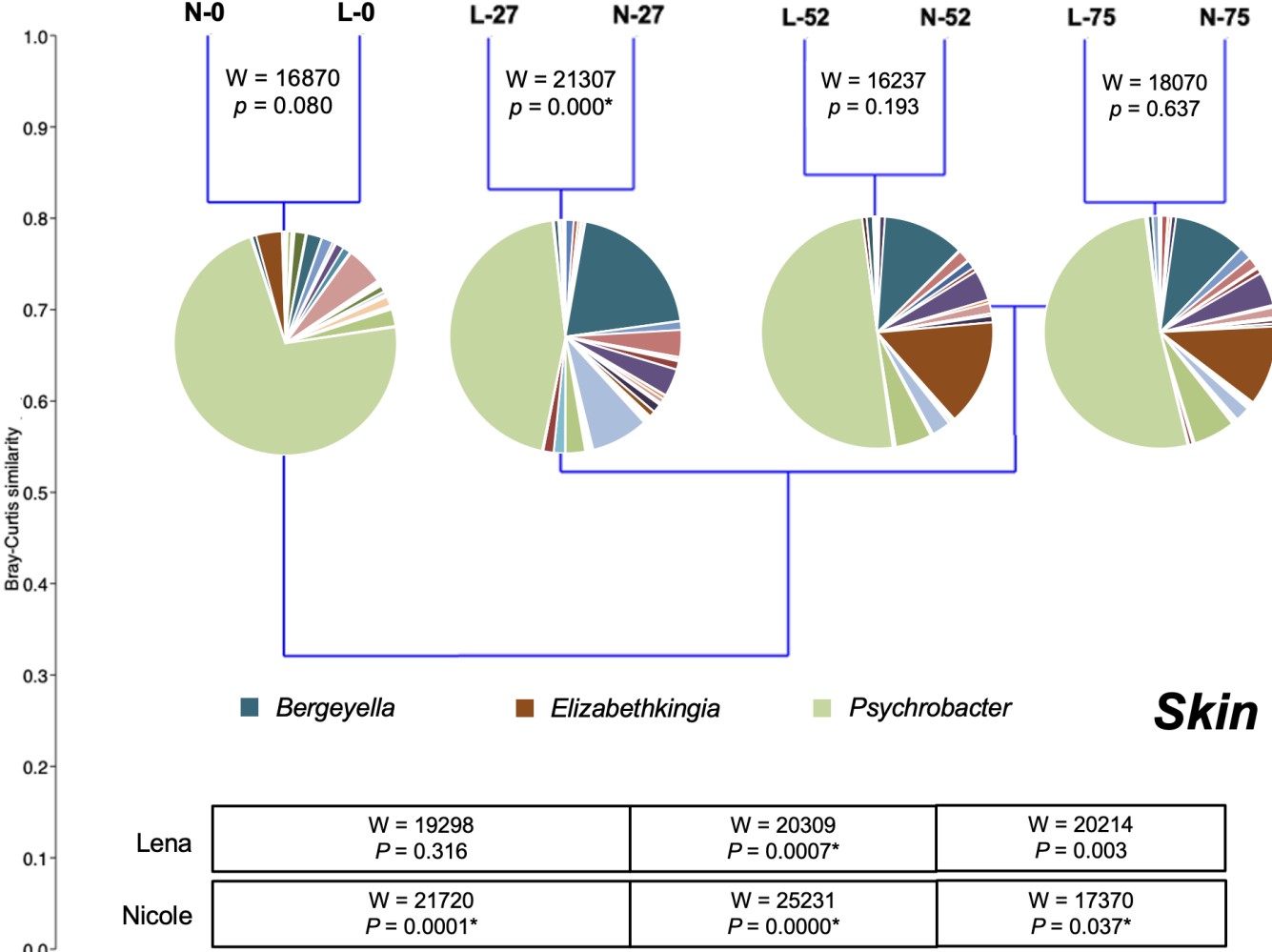

**FIG 4** Cluster analysis of the bacterial operational taxonomic unit abundances on the skin of two hospitalized *Monachus monachus* pups and statistical differences (* indicates $P < 0.05$) between consecutive sampling dates. L, Lena; N, Nicole. L/N numbers indicate the day of sampling. Abundances in the pie charts represent the average relative abundance of bacterial genera within each cluster of samples.

of microbiomes is now well accepted for both wild and domesticated or captive animals (2, 3). In addition, animal microbiomes are nowadays considered to hold important roles in the conservation of their hosts (13). For captive animals who need special care for health or rehabilitation reasons, knowing their microbiome under various environmental conditions or health statuses and how this is shaped, is becoming more important for such practices [e.g., (36, 37)].

A large part of the conservation plan for the Mediterranean monk seal includes rescuing, rehabilitation, and release of stranded sick or injured adult seals or orphan seal pups by the MOm and the School of Veterinary Medicine, Aristotle University of Thessaloniki, Greece. However, to date, there are no microbiome data for this species, either in natural or captive animals. Limited knowledge exists for the closely related species of the Hawaiian monk seal (*Monachus/Neomonachus schauinslandi*) regarding the presence of antibodies to specific pathogenic viruses and bacteria (30), its oral and nasal aerobic bacteria (31), and some cultivable bacterial species related to animal health (32). However, the gut and skin microbiomes are now recognized to be central for the nutrition and health of marine mammals, as for all animals (18, 38). In the present study, we investigated for the first time the succession of fecal and skin bacterial communities of two rescued female Mediterranean monk seal *Monachus monachus* pups during their rehabilitation period prior to their release at sea.

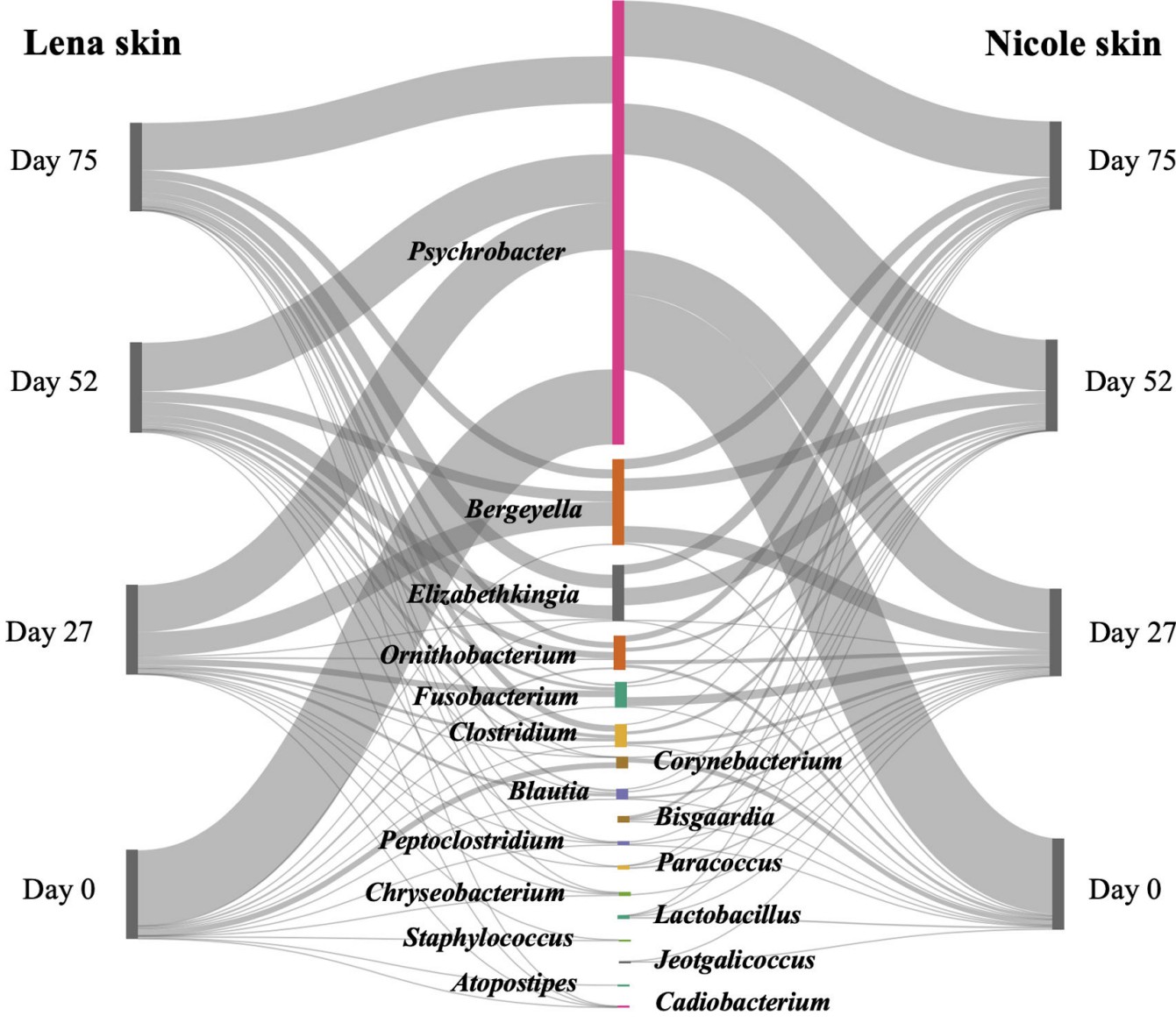

**FIG 5** Dominant (≥80% relative abundance) bacterial genera found on the skin of two hospitalized *Monachus monachus* pups in each sampling point.

The two individuals had similar bacterial profiles both in their skin and fecal microbiota, based on their statistical differences and the high number of shared OTUs, most likely due to their "common garden" (39, 40) environmental conditions, as both animals lived in the same water tank and had the same diet and health care practices. However, in each individual, their skin and fecal microbiota were different (Table 1). Such separation of the two bacterial communities has also been reported in spotted seals (41) as the gastrointestinal tract and skin select for different bacteria suggesting that their microbiota are shaped by different factors and, possibly, provide different services to their hosts.

### Fecal microbiota

The succession of the fecal bacterial microbiota was clustered in two distinct periods being ca. 40% similar to each other (Fig. 2). The first period represents, at least partially, the natural fecal microbiota of the pups, since during this period, no medication or probiotics were given; the impact of the provided nutritional supplements is more likely to be detected in the second period. In the first period, the fecal microbiota was

dominated by the *Shigella*, *Streptococcus*, *Enterococcus*, *Lactobacillus,* and *Escherichia* genera. *Shigella* and *Escherichia* have been found to co-occur in the feces of captive belugas and Pacific white-sided dolphins (42), dwarf sperm whale (*Kogia breviceps*) (43), and harbor seals (44). However, it is often considered as a disease-causative agent for humans and other primates, but it remains unclear if it is pathogenic to other mammals, including marine ones. *Escherichia coli* 0157 strains have been isolated from captured wild *Neomonachus schauinslandi* adult individuals, a closely related species to *M. monachus*, although the animals were not diseased, and have been hypothesized that these, along with other human-related microorganisms, are found in the seals due to their close living in human-dominated ecosystems (45). The evidence for the occurrence of antibiotic-resistant bacteria in marine mammals is accumulating and concerns several of these animal species (46–49). Cultured *E. coli* prevailed in the feces of stranded—but not diseased—harbor seal (*Phoca vitulina*) individuals at admission to rehabilitation compared to wild-caught ones (50), and this is a point of concern during rehabilitation, i.e., the transfer of microorganisms from humans to the hospitalized animals. *Streptococcus* is another potentially detrimental group of bacteria, but it has not been directly related to pathogenesis in marine mammals; it has been found in the oral cavity and intestine of the Yangtze finless porpoise (*Neophocaena asiaeorientalis*) (51) and sperm whales (*Physeter catodon*) (52). *Enterococcus* has been reported to be among the dominant fecal bacteria of captive or stranded but not diseased cetaceans (42, 52–56). In the present study, whether detrimental or not, the considerable decrease of *Shigella* (from 35.2% to 4.4%), *Escherichia* (from 5.5% to 0.7%), and *Streptococcus* (from 17.1% to 0.1%) in the second period after metronidazole admission reduced any potential risks from these bacteria. *Lactobacillus* was the last dominant group of the first period and is a well-known beneficial microorganism of the gastrointestinal tract; indeed, *Lactobacillus* strains with probiotic metabolic features have been isolated from the bottlenose dolphin (*Tursiops truncatus*) (57).

In the second—and longest—period (days 29–66 and 25–64 for Lena and Nicole, respectively) of the rehabilitation process, a major turnover in the dominant OTUs was observed, dominated by the *Clostridium, Blautia, Fusobacterium, Edwardsiella,* and *Bacteroides* genera. The top dominant OTUs were related to *Clostridium* with collective 44.5% relative abundance throughout this period. The genus *Clostridium* could be considered as a resident member of the pups' microbiota, as it very frequently occurs among the most dominant bacteria found in gut and other tissues of healthy, captive, stranded, and dead marine mammals [reference (58) and references therein]. The genus contains pathogenic species, as well, with *Clostridium perfringens* being the most frequent pathogen found in several marine mammals (59–62). In the present study, the *Clostridium*-related sequences cannot be affiliated to any of the known species of this group, but since the pups were not diseased, we hypothesize that these OTUs do not represent pathogenic members of this genus, rendering these specific bacteria as commensals, if not beneficial, for the *M. monachus* pups.

To date, there are no reports on the occurrence of *Blautia* in marine mammal gut or feces. This genus, which is commonly found in terrestrial mammals, has been recently suggested to hold beneficial metabolic traits for their hosts (63), and its occurrence as the second most dominant group in the second rehabilitation period is rather desired. The genus *Fusobacterium* had 10.3% relative abundance in this period. This genus has been found in the oral cavity (51), the genitals (64), and the fecal material of other cetaceans (52, 65), as well. The phylum Fusobacteria has been found to be characteristic of marine carnivores when compared to terrestrial carnivores (29). Although it is considered to include potential pathogens to humans and animals (52, 66), in the present study, its high abundance was not related to any pathogenies in the two pups. *Edwardsiella* has been associated with diseased cetaceans (67) while pathogenic species, like *Edwardsiella tarda*, have been isolated from non-diseased animals (48, 68, 69). This bacterium could be a threat for the released pups, as its abundance seemed to increase toward the end of the rehabilitation period. Of similar abundance was the genus

*Bacteroides* whose abundance increased from 0.1% to 5.3% between the first and second periods. This genus has been found in the oral (51) and fecal (53, 54, 70) microbiome of cetaceans. In humans, it is considered one of the commensal bacteria that first colonize the gut after vaginal birth (71), and this could be also the case for the *M. monachus* pups.

The overall care system resulted in a very good macroscopic condition of the two pups prior to their release and rather stable fecal bacterial communities as suggested by the robustness index, although Lena's fecal bacterial microbiota at the last sampling was slightly less stable compared to the rest of the samplings. The antibiotic's impact on the fecal microbiota eliminated the potentially detrimental *Shigella* dominance in the first period. The probiotics' impact on the fecal microbiota of the pups could not be evaluated in the present study as it was administered on days 72 and 64 for Lena and Nicole, respectively. One of the two pups was photographed (identified by the presence of a marker tag) in a healthy state circa 6 months after release in a regularly monitored seal cave.

## Skin

The skin bacterial microbiota showed a converging pattern from the beginning to the end of the rehabilitation period, as a result of adaption to the artificial environment of the rehabilitation water tank. No statistical differences between the two pups were observed except on day 27. These final skin bacterial communities, prior to the release of the animals, seem to be more stable than the initial ones, as assessed by the robustness index. The observed statistically significant differences in all but one case between consecutive samplings for each pup (Fig. 4) are attributed to the changes in the relative abundance of the same OTUs as the overlap in OTU occurrence was high (Fig. S1).

The three dominant (Fig. 4 and 5) genera *Psychrobacter, Elizabethkingia,* and *Bergeyella* showed different patterns during the rehabilitation. Although *Psychrobacter*-related OTUs dominated in each sampling, their relative abundance decreased gradually during the rehabilitation period. This genus is common on the skin of marine mammals (51, 72–74), reflecting its ubiquity in the marine environment (75, 76). Although it remains unclear if some of its species could be true pathogens, an extensive comparative genomics study concluded that the genus's strains belong to either the flexible ecotype, which can grow at warm temperatures, and so, it can colonize mammalian skin and other tissues, or the restricted ecotype, i.e., the pure psychrophiles, free-living, and general-ist strains found in the world ocean. Both ecotypes have a pathobiont evolutionary origin, whose virulence was lost or weakened via genome reduction (77). Its dominance from the pups' transition from the natural marine to the artificial environment of the rehabilitation water tanks is in accordance with its high adaptability (75, 77).

In our study, the *Elizabethkingia*-related bacteria were temporarily favored during the rehabilitation period, but at the last sampling, they decreased to <0.1%. Although, to date, no reports exist on the occurrence of *Elizabethkingia* in marine mammals, these microorganisms are of special interest as some of its species have been fish-associated and reported as pathogenic or spoilage microorganisms (78). Moreover, infections of *Elizabethkingia meningoseptica* are considered serious and dangerous in humans (79), cows (80), and frogs (81). It is possible that the provided metronidazole in the two seal pups controlled the uprising of the *Elizabethkingia*-related bacteria toward the last half of their rehabilitation period.

*Bergeyella*-related bacteria are another group of skin-associated microorganisms that were favored under the rehabilitation conditions. Its dominance increased between the first and the last sampling, possibly not affected by the provided antibiotic. Although there are no reports on *Bergeyella* in seals, members of this genus have been found with higher abundance in bottlenose dolphin (*Tursiops truncatus*) calves compared to adult males (82). Although in the present study the species identification is of limited security due to the inherent decreased predictability of the short sequence lengths of high-throughput sequencing, some species of this genus could pose a potential risk to

the pups, such as *Bergeyella zoohelcum,* which is a pathogen of the upper respiratory tract of dogs, cats, and other mammals (83).

For the successful reintroduction processes of protected animal species, such as the Mediterranean monk seal (*Monachus monachus*), the final aim of any rehabilitation process is to have healthy animals prior to their release in the wild. Both detrimental and beneficial microorganisms associated with these animals are of central importance for the reintroduction to the wild, with housing and care conditions being at the frontline (84). The present study investigated, for the first time, the skin and fecal microbiota succession of two rescued female *M. monachus* pups during their rehabilitation period. It revealed very low individual variability in both skin and fecal microbiota and some dominant bacterial genera that have been reported for the first time in *M. monachus* or even marine mammals in general. The forecasting power of microbiomes (85) along with its now fast-advancing microbiome engineering (86, 87) even for conservation (13) and halting biodiversity loss (7) opens the way for moving from observational to interventional and functional microbiome research for the protected Mediterranean monk seal.

## MATERIALS AND METHODS

### Rehabilitation

Two female *M. monachus* pups, named as Lena and Nicole, were admitted to the MOm– Monk Seal Rehabilitation Center, Attiki, Greece, on 24 September and 10 October 2019, respectively. The pups were found stranded in the eastern Peloponnese (Lena) and Northeast Euboea (Nicole) on 23 September 2019 (Lena) and 09 October 2019 (Nicole) and weighed 15.9 kg (Lena) and 16.5 kg (Nicole). The estimated pups' ages were 7 and 20 d for Lena and Nicole, respectively. The animals spent 130 (Lena) and 113 (Nicole) d under rehabilitation conditions before being released back into the wild. The seal holding area included a dry platform (ca. 9 $m^2$) and a sea water tank (ca. 10 $m^3$). The recirculating tank water temperature was kept at 17 ± 2°C. The sea water was brought in from the nearby coast of Artemida, Attiki, by a tanker truck and was renewed on average every 6 d. This water was treated continuously by a protein skimmer and every 2–3 d with controlled chlorine doses. The tank water was monitored visually daily, and the origin water was tested monthly. The pups were fed with *Scomber scomber,* which was provided as fish porridge (up to days 49 and 40 for Lena and Nicole, respectively), a combination of fish porridge and whole fish (between days 49–79 and 40–61 for Lena and Nicole, respectively), and for the rest of the rehabilitation time as whole fish. The feeding frequency ranged from one to six times per day, depending on the growth stage of the pups. On their release date, the pups were clinically healthy and weighed 57.4 kg (Lena) and 55.0 kg (Nicole).

To prevent dehydration, an electrolyte solution of Almora sachet PLUS (Elpen, Greece) was provided from the start alone for the first 48 hours and then in fish porridge up to days 79 (Lena) and 61 (Nicole). Boiled Quaker oat extract was provided for the first 8 d (Lena) and 5 d (Nicole). One Aquavits tablet (International Zoo Veterinary Group, UK) was given once daily between days 6 and 109 for Lena and days 12 and 91 for Nicole. Pet-Cal (Zoetis, USA) was used as a supplement of phosphorous, calcium, and vitamin D3, given as 1.5–5 tablets/d from day 6 (Lena) and day 12 (Nicole) until the end of their rehabilitation period. Lena was given daily dosages of 4–20 mL of Salmopet salmon oil (MarinPet, Norway) on days 5–17 and 30–41 and Nicole 5–15 mL on days 16–23. Finally, one sachet of the probiotics supplement Purina Pro Plan Canine FortiFlora (Purina, USA) was given to Lena daily between days 72 and 82 and then every 4 d up to day 125. Nicole got the same probiotics supplement with the same dosage between days 53 and 61, and after that, she was receiving a single sachet every 3 or 4 d until day 111.

Metronidazole was given as prophylaxis to both pups. Lena received daily 5 mg/kg body weight of metronidazole between days 39 and 70. The respective

daily dosage for Nicole was the same between days 20 and 50. Nicole was treated twice a day with azithromycin eye drops (Azyter; Laboratoires Thea, France) from days 6–17 to days 29–30.

## Molecular analyses and data processing

A total of nine individual fecal and four skin swab samples were collected and analyzed for each individual during their rehabilitation period. Our first sampling (day 0) corresponds to the second day of the pups in the rehabilitation center. Fecal samples were collected immediately upon defecation from each individual. Pre-sterilized cotton swab scrapings were retrieved along the sides of each individual. All samples were immediately frozen at −20°C and then at −80°C. Bulk DNA was extracted from ca. 0.3 g of fecal material or the whole cotton swab using the NucleoSpin Soil DNA extraction kit (Macherey-Nagel, Germany) according to the manufacturer's guidelines.

For the PCR amplification of the V3–V4 regions of the bacterial 16S rRNA gene from the bulk extracted DNA, we used the primer pair S-D-Bact-0341-b-S-17 and S-D-Bact-115 0785 a-A-21 (88). The amplified sequences were sequenced on a MiSeq Illumina instrument (2 × 300 bp) at the MRDNA Ltd. (Shallowater, TX, USA) sequencing facilities. Unprocessed DNA sequences are available in the Sequence Read Archive (https://www.ncbi.nlm.nih.gov/sra/) under BioSample SAMN32536541 of the BioProject PRJNA917309. All processing of the raw 16S rRNA gene sequences was performed by using the MOTHUR standard operating procedure (v.1.46.1) (89, 90). The resulting operational taxonomic units were grouped as identical at the 97% cut-off similarity level and were classified with the SILVA database release 138 (91, 92). For those OTUs that were designated as "unaffiliated," their closest relatives were found by using Nucleotide Blast (http://blast.ncbi.nlm.nih.gov).

Data and statistical analysis and graphic illustrations were performed using Palaeontological STudies software (93) and the vegan package (94) in R Studio platform version 1.1.419 (95) with 3.4.3 R version. We applied cluster analysis based on the unweighted pair group method with arithmetic mean Bray–Curtis similarity. PERMANOVA was used to detect differences between the fecal and skin bacterial microbiota of the two pups, and the Wilcoxon test was applied to detect the microbiota differences between sequential samplings. The robustness of the fecal and skin bacterial microbiota to temporal stability and resistance were investigated in this study. Robustness is an index that assesses the degree of a community's structural constancy over time (96), its ability to resist change following perturbation (97, 98), and its resilience, i.e., its ability to return to an initial structure following perturbation (99).

## ACKNOWLEDGMENTS

We would like to thank the Attica Zoological Park for the logistical and financial support of the two orphan monk seal pups. The rehabilitation of the two pups was conducted following strict protocols and with all necessary permits from the national relevant authorities of Greece.

K.K. and A.D. conceived the ideas, designed the methodology, and led the writing of the manuscript; A.D., A.M., and K.K. performed microbiota and bioinformatics and data analysis; A.D., K.K., A.K., E.T., and P.D. performed all sampling and animal care during rehabilitation. A.K. was responsible for the veterinary care of the animals. All authors contributed critically to the drafts and gave final approval for publication.

## AUTHOR AFFILIATIONS

[1]Department of Ichthyology and Aquatic Environment, University of Thessaly, Volos, Greece
[2]MOm/Hellenic Society for the Study and Protection of the Monk Seal, Athens, Greece
[3]School of Veterinary Medicine, Faculty of Health Sciences, Aristotle University of Thessaloniki, Thessaloniki, Greece

## PRESENT ADDRESS

Alexandra Meziti, Department of Marine Sciences, University of the Aegean, University Hill, Mytilene, Greece

## AUTHOR ORCIDs

Aggeliki Dosi ⓘ http://orcid.org/0009-0004-5800-8627
Alexandra Meziti ⓘ http://orcid.org/0000-0001-5672-7653
Kimon Koemtzopoulos ⓘ http://orcid.org/0009-0009-0705-2712
Anastasia Komnenou ⓘ http://orcid.org/0000-0001-6715-3798
Panagiotis Dendrinos ⓘ http://orcid.org/0000-0002-2499-7058
Konstantinos Kormas ⓘ http://orcid.org/0000-0002-8214-0175

## AUTHOR CONTRIBUTIONS

Aggeliki Dosi, Conceptualization, Formal analysis, Methodology, Validation, Writing – original draft | Alexandra Meziti, Formal analysis, Investigation, Methodology, Validation, Writing – review and editing | Eleni Tounta, Investigation, Methodology, Writing – review and editing | Kimon Koemtzopoulos, Investigation, Methodology, Writing – review and editing | Anastasia Komnenou, Investigation, Methodology, Writing – review and editing | Panagiotis Dendrinos, Investigation, Methodology, Writing – review and editing | Konstantinos Kormas, Conceptualization, Data curation, Investigation, Methodology, Project administration, Resources, Supervision, Validation, Writing – original draft

## ADDITIONAL FILES

The following material is available online.

### Supplemental Material

**Tables S1 and S2, Figures S1 and S2 (Spectrum02805-23-s0001.docx).** Supplementary material.

### Open Peer Review

**PEER REVIEW HISTORY (review-history.pdf).** An accounting of the reviewer comments and feedback.

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
