## [Reviewer comments · Microbiology Spectrum]

Microbiology Spectrum

Fecal and skin microbiota of two rescued Mediterranean monk seal pups during rehabilitation

Aggeliki Dosi, Alexandra Meziti, Eleni Tounta, Kimon Koemtzopoulos, Anastasia Komnenou, Panagiotis Dendrinou, and Konstantinos Kormas

Corresponding Author(s): Konstantinos Kormas, Panepistemio Thessalias Tmema Geoponias Ichthyologias kai Ydatinou Periballontos

Review Timeline:

Submission Date:	July 9, 2023
Editorial Decision:	September 19, 2023
Revision Received:	October 15, 2023
Editorial Decision:	November 10, 2023
Revision Received:	November 13, 2023
Accepted:	November 15, 2023

Editor: Bernadette Connors

Reviewer(s): The reviewers have opted to remain anonymous.

Transaction Report:

DOI: <https://doi.org/10.1128/spectrum.02805-23>

September 19, 2023

Dr. Konstantinos Aristomenis Kormas
Panepistemio Thessalias Tmema Geoponias Ichthyologias kai Ydatinou Periballontos
Ichthyology & Aquatic Environment
Fitoko
Volos 38446
Greece

Re: Spectrum02805-23 (Fecal and skin microbiota of two rescued Mediterranean monk seal pups during rehabilitation)

Dear Dr. Konstantinos Aristomenis Kormas:

Link Not Available

Sincerely,

Bernadette Connors

Journals Department
Reviewer comments:

Reviewer #1 (Public repository details (Required)):

BioSample and BioProject accession codes return "No items found" from SRA. Please see attached photos.

Reviewer #1 (Comments for the Author):

The investigation provides much needed data on microbiota of marine mammals which are difficult to sample in the wild. The protocols done are acceptable/standard, and analyses robust enough for the research questions.

Reviewer #2 (Comments for the Author):

Dosi et al. investigated the skin and gut bacterial communities in two seal pups during their rehabilitation period. This paper is notably well-written. However, a primary concern arises from the relatively small sample size, as data was collected from just two seal pups. Nonetheless, considering the inherent challenges in acquiring such precious samples, the author's effort in devising a longitudinal sampling plan is commendable. Overall, this study offers valuable insights into the microbiome associated with seal pups.

I have a couple of questions regarding the statistical aspects of the study:

1. Lines 138-140 and 386-389 were pairwise PERMANOVA analyses conducted on the fecal and skin bacterial communities of the two seal pups?
2. Regarding Figures 2 and 4, do the pie charts represent the average relative abundance of bacterial genera (those exceeding 5% abundance) within each cluster of samples? If so, clarify this in the figure legends.
3. For Figure 2, specifically lines 388-389, you aimed to compare differences in community composition between two consecutive sampling points, such as L-0 vs. L-11 for Lena. It seems that only one sample was collected at each sampling point for each pup. Please explain how the Wilcoxon test was applied to compare two samples in this context.

Staff Comments:

Preparing Revision Guidelines

Please return the manuscript within 60 days; if you cannot complete the modification within this time period, please contact me. If you do not wish to modify the manuscript and prefer to submit it to another journal, please notify me of your decision immediately so that the manuscript may be formally withdrawn from consideration by Microbiology Spectrum.

**Fecal and skin microbiota of two rescued Mediterranean monk seal pups during**
**rehabilitation**

Aggeliki Dosi^a, Alexandra Meziti^{a,#}, Eleni Tounta^b, Kimon Koentzopoulos^b, Anastasia
Komnenou^c, Panagiotis Dendrinou^b, Konstantinos Kormas^{a,1}

9 ^a Department of Ichthyology and Aquatic Environment, University of Thessaly, 384 46
Volos, Greece.

12 ^b MOM/Hellenic Society for the Study and Protection of the Monk Seal, Solomou 18, 10682,
Athens, Greece.

15 ^c School of Veterinary Medicine, Faculty of Health Sciences, Aristotle University of
16 Thessaloniki, 574 00 Thessaloniki, Greece.

[#] Present address: Department of Marine Sciences, University of the Aegean, University Hill,
81100 Mytilene, Greece

¹ Address correspondence to Konstantinos Kormas, kkormas@uth.gr

Running title: Rescued *Monachus monachus* skin and fecal microbiota

[revised manuscript text omitted]

Greece) was provided from start alone the first 48 hours and then and in fish porridge to day
79 (Lena) and 61 (Nicole). Boiled Quaker oats extract was provided for the first 8 (Lena) and
5 (Nicole) days. One Aquavits tablet (International Zoo Veterinary Group, UK) was given
once daily between days 6 and 109 for Lena and days 12 and 91 for Nicole. PetCal (Zoetis,
USA) was used as a supplement of phosphorous, calcium, and vitamin D3, given as 1.5-5
tablets/d from day 6 (Lena) and day 12 (Nicole) until the end of their rehabilitation period.
Lena was given daily dosages of 4-20 ml of SalmoPet salmon oil (MarinPet, Norway) in days
5-17 and 30-41 and Nicole 5-15 ml in days 16-23. Finally, one sachet of the probiotics
supplement PURINA PRO PLAN Canine FortiFlora (Purina, USA) was given to Lena daily
between days 72-82 and then every four days up to day 125. Nicole got the same probiotics
supplement with the same dosage between days 53-61 and after that she was receiving a
single sachet every three or four days until day 111.

Metronidazole was given as prophylaxis to both pups. Lena received daily 5mg/kg
body weight of metronidazole between days 39-70. The respective daily dosage for Nicole
was the same between days 20-50. Nicole was treated twice a day with azithromycin eye
drops (Azyter; Laboratoires Thea, France) from days 6-17 and days 29-30.

**Molecular analyses and data processing.** A total of nine individual fecal and four
skin swab samples were collected and analyzed for each individual during their
rehabilitations period. Our first sampling (day 0) corresponds to the second day of the pups in
the rehabilitation center. Fecal samples were collected immediately upon defecation from
each individual. Pre-sterilized cotton swab scrapings were retrieved along the sides of each
individual. All samples were immediately frozen at -20⁰C and then at -80⁰C. Bulk DNA was
extracted from ca. 0.3 g of fecal material or the whole cotton swab using the NucleoSpin Soil
DNA extraction kit (Machery-Nagel, Germany) according to the manufacturer's guidelines.

For the PCR amplification of the V3–V4 regions of the bacterial 16S rRNA gene
from the bulk extracted DNA, we used the primer pair S-D-Bact-0341-b-S-17 and S-D-Bact-
115 0785-a-A-21 (88). The amplified sequences were sequenced on a MiSeq Illumina
instrument (2x300 bp) at the MRDNA Ltd. (Shallowater, TX, USA) sequencing facilities.
Unprocessed DNA sequences are available in the Sequence Read Archive
(<https://www.ncbi.nlm.nih.gov/sra/>) under BioSample SAMN32536541 of the BioProject
PRJNA917309. All processing of the raw 16S rRNA gene sequences was performed by using
the MOTHUR standard operating procedure (v.1.46.1) (89, 90). The resulting operational
taxonomic units (OTUs) were grouped as identical at 97% cut-off similarity level and were
classified with the SILVA database release 138 (91, 92). For those OTUs which were
designated as “unaffiliated”, their closest relatives were found by using Nucleotide Blast
(<http://blast.ncbi.nlm.nih.gov>).

Data and statistical analysis and graphic illustrations were performed using
Palaeontological STudies (PAST) software (93) and the vegan package (94) in R Studio
platform Version 1.1.419 (95) with 3.4.3 R version. We applied cluster analysis based on the
unweighted pair group method with arithmetic mean Bray-Curtis similarity. Permutational
multivariate analysis of variance (PERMANOVA) was used to detect differences between the
fecal and skin bacterial microbiota of the two pups and the Wilcoxon test was applied to
detect the microbiota differences between sequential samplings. The robustness of the fecal
and skin bacterial microbiota to temporal stability and resistance were investigated in this
study. Robustness is an index which assesses the degree of a community's structural

constancy over time (96), its ability to resist change following perturbation (97, 98), and its
resilience, i.e., its ability to return to an initial structure following perturbation (99).

**ATTACHED PHOTOS**

**ACKNOWLEDGEMENTS**

We would like to thank the Attica Zoological Park for the logistical and financial
support of the two orphan monk seal pups. The rehabilitation of the two pups was conducted
following strict protocols and with all necessary permits from the national relevant authorities
of Greece.

**CONFLICT OF INTEREST**

The authors declare no conflict of interest.

**AUTHOR CONTRIBUTIONS**

Kon. K. and A.D. conceived the ideas, designed methodology and led the writing of
the manuscript; A.D., A.M. and Kon. K. performed microbiota and bioinformatics and data
analysis; A.D., Kim. K., A.K., E.T. and P.D. performed all sampling and animal care during
rehabilitation. A.K. was responsible for the veterinary care of the animals. All authors
contributed critically to the drafts and gave final approval for publication.

**Table 1.** PERMANOVA results of the fecal and skin bacterial operational taxonomic units in
the feces and skin of two hospitalized *Monachus monachus* pups. Upper half are the p values
and lower half of the table are the F values. Star indicates $p < 0.05$.

[revised manuscript text omitted]

Dear Editor,

We thank you and the reviewers for your time and effort spent on our manuscript. We provide below our point-by-point response (in black letters) in all of the reviewers' comments (in red letters).

Reviewer #1 (Public repository details (Required)):

BioSample and BioProject accession codes return "No items found" from SRA. Please see attached photos.

We have requested from NVBI to release the dataset upon publication of the paper. We provide below the "Create review link":

<https://dataview.ncbi.nlm.nih.gov/object/PRJNA917309?reviewer=mq7v3cr6vc2phlk8kfu2e4oo59>

Reviewer #1 (Comments for the Author):

The investigation provides much needed data on microbiota of marine mammals which are difficult to sample in the wild. The protocols done are acceptable/standard, and analyses robust enough for the research questions.

Reviewer #2 (Comments for the Author):

Dosi et al. investigated the skin and gut bacterial communities in two seal pups during their rehabilitation period. This paper is notably well-written. However, a primary concern arises from the relatively small sample size, as data was collected from just two seal pups.

Nonetheless, considering the inherent challenges in acquiring such precious samples, the author's effort in devising a longitudinal sampling plan is commendable. Overall, this study offers valuable insights into the microbiome associated with seal pups.

We thank both the reviewers for their positive comment on the merit of our paper.

I have a couple of questions regarding the statistical aspects of the study:

1. Lines 138-140 and 386-389 were pairwise PERMANOVA analyses conducted on the fecal and skin bacterial communities of the two seal pups?

We clarified this sentence by rephrasing it to: "PERMANOVA showed that for each pup, skin and fecal bacterial microbiota were significantly different, but no such differences occurred between the two pups' skin and fecal microbiota (Table 1).".

2. Regarding Figures 2 and 4, do the pie charts represent the average relative abundance of bacterial genera (those exceeding 5% abundance) within each cluster of samples? If so, clarify this in the figure legends.

Yes. We have added the following sentence in Figures' 2 and 4 legends: "Abundances in the pie charts represent the average relative abundance of bacterial genera within each cluster of samples."

3. For Figure 2, specifically lines 388-389, you aimed to compare differences in community composition between two consecutive sampling points, such as L-0 vs. L-11 for Lena. It seems that only one sample was collected at each sampling point for each pup. Please explain how the Wilcoxon test was applied to compare two samples in this context.

The Wilcoxon test is used to compare 2 populations with matched samples. We compared the OTUs abundances in each pair of consecutive samples.

Re: Spectrum02805-23R1 (Fecal and skin microbiota of two rescued Mediterranean monk seal pups during rehabilitation)

Dear Dr. Konstantinos Aristomenis Kormas:

Thank you for the privilege of reviewing your work. Below you will find my comments, instructions from the Spectrum editorial office, and the reviewer comments.

Please look at the attached document. One of the reviewer's embedded their comments in the document itself (they are typed in a different style and size of font). I understand that you have already addressed the release of SRA data, but there are a couple of other comments in the document to address.

Revision Guidelines

Sincerely,
Bernadette Connors
Editor
Microbiology Spectrum

Reviewer #1 (Comments for the Author):

During my last evaluation, I have noted that the comments are written on the doc file that I sent. However, I did not see any explanation, comment or answer addressing the issues that I raised, hence, sending the doc file again with tracked edits and comments. Please address my comments.

Reviewer #2 (Comments for the Author):

The authors have addressed all my questions.

**Fecal and skin microbiota of two rescued Mediterranean monk seal pups during**
**rehabilitation**

Aggeliki Dosi^a, Alexandra Meziti^{a,#}, Eleni Tounta^b, Kimon Koentzopoulos^b, Anastasia
Komnenou^c, Panagiotis Dendrinou^b, Konstantinos Kormas^{a,1}

9 ^a Department of Ichthyology and Aquatic Environment, University of Thessaly, 384 46
Volos, Greece.

12 ^b MOM/Hellenic Society for the Study and Protection of the Monk Seal, Solomou 18, 10682,
Athens, Greece.

15 ^c School of Veterinary Medicine, Faculty of Health Sciences, Aristotle University of
16 Thessaloniki, 574 00 Thessaloniki, Greece.

[#] Present address: Department of Marine Sciences, University of the Aegean, University Hill,
81100 Mytilene, Greece

¹ Address correspondence to Konstantinos Kormas, kkormas@uth.gr

Running title: Rescued *Monachus monachus* skin and fecal microbiota

[revised manuscript text omitted]

Greece) was provided from start alone the first 48 hours and then and in fish porridge to day
79 (Lena) and 61 (Nicole). Boiled Quaker oats extract was provided for the first 8 (Lena) and
5 (Nicole) days. One Aquavits tablet (International Zoo Veterinary Group, UK) was given
once daily between days 6 and 109 for Lena and days 12 and 91 for Nicole. PetCal (Zoetis,
USA) was used as a supplement of phosphorous, calcium, and vitamin D3, given as 1.5-5
tablets/d from day 6 (Lena) and day 12 (Nicole) until the end of their rehabilitation period.
Lena was given daily dosages of 4-20 ml of SalmoPet salmon oil (MarinPet, Norway) in days
5-17 and 30-41 and Nicole 5-15 ml in days 16-23. Finally, one sachet of the probiotics
supplement PURINA PRO PLAN Canine FortiFlora (Purina, USA) was given to Lena daily
between days 72-82 and then every four days up to day 125. Nicole got the same probiotics
supplement with the same dosage between days 53-61 and after that she was receiving a
single sachet every three or four days until day 111.

Metronidazole was given as prophylaxis to both pups. Lena received daily 5mg/kg
body weight of metronidazole between days 39-70. The respective daily dosage for Nicole
was the same between days 20-50. Nicole was treated twice a day with azithromycin eye
drops (Azyter; Laboratoires Thea, France) from days 6-17 and days 29-30.

**Molecular analyses and data processing.** A total of nine individual fecal and four
skin swab samples were collected and analyzed for each individual during their
rehabilitations period. Our first sampling (day 0) corresponds to the second day of the pups in
the rehabilitation center. Fecal samples were collected immediately upon defecation from
each individual. Pre-sterilized cotton swab scrapings were retrieved along the sides of each
individual. All samples were immediately frozen at -20⁰C and then at -80⁰C. Bulk DNA was
extracted from ca. 0.3 g of fecal material or the whole cotton swab using the NucleoSpin Soil
DNA extraction kit (Machery-Nagel, Germany) according to the manufacturer's guidelines.

For the PCR amplification of the V3–V4 regions of the bacterial 16S rRNA gene
from the bulk extracted DNA, we used the primer pair S-D-Bact-0341-b-S-17 and S-D-Bact-
115 0785-a-A-21 (88). The amplified sequences were sequenced on a MiSeq Illumina
instrument (2x300 bp) at the MRDNA Ltd. (Shallowater, TX, USA) sequencing facilities.
Unprocessed DNA sequences are available in the Sequence Read Archive
(<https://www.ncbi.nlm.nih.gov/sra/>) under BioSample SAMN32536541 of the BioProject
PRJNA917309. All processing of the raw 16S rRNA gene sequences was performed by using
the MOTHUR standard operating procedure (v.1.46.1) (89, 90). The resulting operational
taxonomic units (OTUs) were grouped as identical at 97% cut-off similarity level and were
classified with the SILVA database release 138 (91, 92). For those OTUs which were
designated as “unaffiliated”, their closest relatives were found by using Nucleotide Blast
(<http://blast.ncbi.nlm.nih.gov>).

Data and statistical analysis and graphic illustrations were performed using
Palaeontological STudies (PAST) software (93) and the vegan package (94) in R Studio
platform Version 1.1.419 (95) with 3.4.3 R version. We applied cluster analysis based on the
unweighted pair group method with arithmetic mean Bray-Curtis similarity. Permutational
multivariate analysis of variance (PERMANOVA) was used to detect differences between the
fecal and skin bacterial microbiota of the two pups and the Wilcoxon test was applied to
detect the microbiota differences between sequential samplings. The robustness of the fecal
and skin bacterial microbiota to temporal stability and resistance were investigated in this
study. Robustness is an index which assesses the degree of a community's structural

constancy over time (96), its ability to resist change following perturbation (97, 98), and its
resilience, i.e., its ability to return to an initial structure following perturbation (99).

**ATTACHED PHOTOS**

**ACKNOWLEDGEMENTS**

We would like to thank the Attica Zoological Park for the logistical and financial
support of the two orphan monk seal pups. The rehabilitation of the two pups was conducted
following strict protocols and with all necessary permits from the national relevant authorities
of Greece.

**CONFLICT OF INTEREST**

The authors declare no conflict of interest.

**AUTHOR CONTRIBUTIONS**

Kon. K. and A.D. conceived the ideas, designed methodology and led the writing of
the manuscript; A.D., A.M. and Kon. K. performed microbiota and bioinformatics and data
analysis; A.D., Kim. K., A.K., E.T. and P.D. performed all sampling and animal care during
rehabilitation. A.K. was responsible for the veterinary care of the animals. All authors
contributed critically to the drafts and gave final approval for publication.

**Table 1.** PERMANOVA results of the fecal and skin bacterial operational taxonomic units in
the feces and skin of two hospitalized *Monachus monachus* pups. Upper half are the p values
and lower half of the table are the F values. Star indicates $p < 0.05$.

[revised manuscript text omitted]

Dear Editor,

We thank you and the reviewers for your time and effort spent on our manuscript. We provide below our point-by-point response (in black letters) in all of the reviewers' comments (in red letters).

Reviewer #1 (Public repository details (Required)):

BioSample and BioProject accession codes return "No items found" from SRA. Please see attached photos.

We have requested from NVBI to release the dataset upon publication of the paper. We provide below the "Create review link":

<https://dataview.ncbi.nlm.nih.gov/object/PRJNA917309?reviewer=mq7v3cr6vc2phlk8kfu2e4oo59>

Reviewer #1 (Comments for the Author):

The investigation provides much needed data on microbiota of marine mammals which are difficult to sample in the wild. The protocols done are acceptable/standard, and analyses robust enough for the research questions.

Reviewer #2 (Comments for the Author):

Dosi et al. investigated the skin and gut bacterial communities in two seal pups during their rehabilitation period. This paper is notably well-written. However, a primary concern arises from the relatively small sample size, as data was collected from just two seal pups.

Nonetheless, considering the inherent challenges in acquiring such precious samples, the author's effort in devising a longitudinal sampling plan is commendable. Overall, this study offers valuable insights into the microbiome associated with seal pups.

We thank both the reviewers for their positive comment on the merit of our paper.

I have a couple of questions regarding the statistical aspects of the study:

1. Lines 138-140 and 386-389 were pairwise PERMANOVA analyses conducted on the fecal and skin bacterial communities of the two seal pups?

We clarified this sentence by rephrasing it to: "PERMANOVA showed that for each pup, skin and fecal bacterial microbiota were significantly different, but no such differences occurred between the two pups' skin and fecal microbiota (Table 1).".

2. Regarding Figures 2 and 4, do the pie charts represent the average relative abundance of bacterial genera (those exceeding 5% abundance) within each cluster of samples? If so, clarify this in the figure legends.

Yes. We have added the following sentence in Figures' 2 and 4 legends: "Abundances in the pie charts represent the average relative abundance of bacterial genera within each cluster of samples."

3. For Figure 2, specifically lines 388-389, you aimed to compare differences in community composition between two consecutive sampling points, such as L-0 vs. L-11 for Lena. It seems that only one sample was collected at each sampling point for each pup. Please explain how the Wilcoxon test was applied to compare two samples in this context.

The Wilcoxon test is used to compare 2 populations with matched samples. We compared the OTUs abundances in each pair of consecutive samples.

Re: Spectrum02805-23R2 (Fecal and skin microbiota of two rescued Mediterranean monk seal pups during rehabilitation)

Dear Dr. Kormas:

Your manuscript has been accepted, and I am forwarding it to the ASM production staff for publication. Your paper will first be checked to make sure all elements meet the technical requirements. ASM staff will contact you if anything needs to be revised before copyediting and production can begin. Otherwise, you will be notified when your proofs are ready to be viewed.

Sincerely,
Bernadette Connors
Editor
Microbiology Spectrum